# *NAGS*, *CPS1*, and *SLC25A13* (Citrin) at the Crossroads of Arginine and Pyrimidines Metabolism in Tumor Cells

**DOI:** 10.3390/ijms24076754

**Published:** 2023-04-04

**Authors:** Melissa Owusu-Ansah, Nikita Guptan, Dylon Alindogan, Michio Morizono, Ljubica Caldovic

**Affiliations:** 1Columbian College of Arts and Sciences, George Washington University, Washington, DC 20052, USA; 2Department of Microbiology, Immunology, and Tropical Medicine, School of Medicine and Health Sciences, George Washington University, Washington, DC 20052, USA; 3School of Mathematics, College of Science and Engineering, University of Minnesota, Minneapolis, MN 55455, USA; 4Department of Genomics and Precision Medicine, School of Medicine and Health Sciences, George Washington University, Washington, DC 20052, USA; 5Center for Genetic Medicine Research, Children’s National Research Institute, Children’s National Hospital, Washington, DC 20010, USA

**Keywords:** N-acetylglutamate synthase, carbamylphosphate synthetase 1, citrin, urea cycle, pyrimidine biosynthesis, CAD complex, tumor, tumorigenesis

## Abstract

Urea cycle enzymes and transporters collectively convert ammonia into urea in the liver. Aberrant overexpression of carbamylphosphate synthetase 1 (*CPS1*) and *SLC25A13* (citrin) genes has been associated with faster proliferation of tumor cells due to metabolic reprogramming that increases the activity of the CAD complex and pyrimidine biosynthesis. N-acetylglutamate (NAG), produced by NAG synthase (NAGS), is an essential activator of CPS1. Although NAGS is expressed in lung cancer derived cell lines, expression of the *NAGS* gene and its product was not evaluated in tumors with aberrant expression of *CPS1* and citrin. We used data mining approaches to identify tumor types that exhibit aberrant overexpression of *NAGS*, *CPS1*, and citrin genes, and evaluated factors that may contribute to increased expression of the three genes and their products in tumors. Median expression of *NAGS*, *CPS1*, and citrin mRNA was higher in glioblastoma multiforme (GBM), glioma, and stomach adenocarcinoma (STAD) samples compared to the matched normal tissue. Median expression of *CPS1* and citrin mRNA was higher in the lung adenocarcinoma (LUAD) sample while expression of *NAGS* mRNA did not differ. High *NAGS* expression was associated with an unfavorable outcome in patients with glioblastoma and GBM. Low *NAGS* expression was associated with an unfavorable outcome in patients with LUAD. Patterns of DNase hypersensitive sites and histone modifications in the upstream regulatory regions of *NAGS*, *CPS1*, and citrin genes were similar in liver tissue, lung tissue, and A549 lung adenocarcinoma cells despite different expression levels of the three genes in the liver and lung. Citrin gene copy numbers correlated with its mRNA expression in glioblastoma, GBM, LUAD, and STAD samples. There was little overlap between *NAGS*, *CPS1,* and citrin sequence variants found in patients with respective deficiencies, tumor samples, and individuals without known rare genetic diseases. The correlation between *NAGS*, *CPS1*, and citrin mRNA expression in the individual glioblastoma, GBM, LUAD, and STAD samples was very weak. These results suggest that the increased cytoplasmic supply of either carbamylphosphate, produced by CPS1, or aspartate may be sufficient to promote tumorigenesis, as well as the need for an alternative explanation of CPS1 activity in the absence of *NAGS* expression and NAG.

## 1. Introduction

The reprogramming of cellular metabolism to favor anabolic pathways can contribute to the uncontrolled growth and proliferation of tumor cells. The metabolic reprogramming that drives tumor development can be achieved through gain-of-function somatic mutations that result in the increased activity of enzymes such as isocitrate dehydrogenase 1 and 2 [1,2,3,4,5]. Heterozygous germline sequence variants that result in a lower activity of fumarate hydratase and succinate dehydrogenase can increase the risk of tumor development [6,7,8]. Aberrant expression of enzymes belonging to catabolic pathways such as urea cycle can contribute to the increased proliferation of tumor cells and unfavorable patient outcomes [9].

The urea cycle comprises six enzymes and two mitochondrial solute carriers that convert ammonia, a product of amino acid catabolism and a potent neurotoxin, into non-toxic urea in the liver [10,11]. Urea cycle enzymes are partitioned between the mitochondria and cytoplasm of liver cells. N-acetylglutamate synthase (NAGS), carbamylphosphate synthetase I (CPSI), and ornithine transcarbamylase (OTC) are located in the mitochondrial matrix [11]. CPSI and OTC collectively convert ammonia, bicarbonate, ATP, and ornithine into citrulline, which is then converted into urea by the cytosolic enzymes argininosuccinate synthetase 1 (ASS1), argininosuccinate lyase (ASL), and arginase 1 (ARG1) [11]. NAGS uses glutamate and acetyl coenzyme A to produce NAG, an obligatory allosteric activator of CPSI [11,12,13,14,15,16,17,18]. NAGS, CPSI, and OTC are also expressed in the small intestine, where they synthesize citrulline which is then transported to the kidneys for the biosynthesis of arginine by ASS1 and ASL [11]. ASS1 and ASL also function in NO signaling, where they catalyze the formation of arginine which is a substrate of NO-synthetase [19]. Mitochondrial solute carriers that function in urea cycle are ornithine transporter 1 (ORNT1), encoded by the *SLC25A15* gene, and aspartate-glutamate carrier 2, also known as citrin or ARALAR 2 and encoded by the *SLC25A13* gene [11,20,21,22]. ORNT1 transports ornithine from the cytoplasm into mitochondria and citrulline out of mitochondria into the cytoplasm [23]. Citrin transports aspartate from mitochondria into the cytoplasm where it is used as an ASS1 substrate [24,25,26]. Citrin also participates in the malate-aspartate shuttle, which maintains the redox balance by transporting electrons produced in the cytoplasm by glycolytic enzymes into mitochondria for use in energy production [22,26]. 

Aberrant expression of urea cycle enzymes and mitochondrial solute carriers can contribute to the metabolic reprogramming of tumor cells [9]. Silencing of ASS1 and ASL in hepatocellular carcinoma, bladder cancer, and lung adenocarcinoma can lead to an increased supply of cytoplasmic aspartate which is a substrate of the CAD complex (carbamylphosphate synthetase 2, aspartate transcarbamylase, and dihydroorotase) that catalyzes the first three reactions of de novo pyrimidine biosynthesis [27,28]. Overexpression of citrin can contribute to the metabolic reprogramming of tumor cells in two ways. First, citrin overexpression can contribute to the increased supply of cytoplasmic aspartate, increased de novo biosynthesis of pyrimidine nucleotides, faster cell proliferation, and unfavorable outcomes in patients with lung adenocarcinoma, bladder cancer, and uterine tumors [22,28]. Second, citrin overexpression can disrupt the function of the malate-aspartate shuttle leading to a redox imbalance and the disruption of energy metabolism in tumor cells [29,30]. Aberrant expression of *CPS1* in tumors that originate in tissues that normally do not express the gene has been associated with an increased cytoplasmic supply of carbamylphosphate (CP), a substrate of the CAD complex, leading to increased de novo pyrimidine biosynthesis, tumor cell proliferation, and unfavorable patient outcomes [31,32,33,34,35]. Although these studies provided evidence of CPS1 enzymatic activity in tumor cells, only one study showed the presence of NAGS in cell lines that model non-small cell lung carcinoma and express CPS1 [36]. Since the aberrant expression of two or more urea cycle genes is needed for the metabolic reprogramming of tumor cells [9], we used data mining approaches to identify tumor types with aberrant expression of *NAGS*, *CPS1,* and citrin genes that can increase the cytoplasmic concentration of CP and aspartate leading to higher CAD activity, biosynthesis of pyrimidine nucleotides, and dysregulation of the cellular pyrimidine–purine balance. We queried repositories of tumor genomic data to determine whether the higher expression of *NAGS*, *CPS1,* and citrin genes correlate with unfavorable patient outcomes. We also investigated molecular mechanisms that could contribute to overexpression of the three genes in tumor cells, assessed differences between somatic and germline sequence variants, and determined if there is a correlation between expression levels of the three genes in individual tumor samples. 

## 2. Results

### 2.1. Aberrant Expression of NAGS, CPS1, and Citrin in Tumors

The Genome Data Analysis Center (GDAC) at the Broad Institute hosts processed gene expression data from The Cancer Genome Atlas (TCGA) Project and makes them available to the research community through the Firehose Data portal [37]. We queried the GDAC Firehose Data Portal for the mRNA expression of urea cycle genes in tumors and matched normal tissues. GDAC uses the Log2 transform of RSEM values [38] to quantify abundance of mRNA. The difference in expression of urea cycle genes between tumor samples and matched normal tissues, expressed as the fold-change of the median log_2_RSEM values, was available for 27 tumor types (Figure 1A). *NAGS* and *CPS1* genes were markedly overexpressed in esophageal carcinoma and kidney chromophobe samples, respectively (Figure 1A). We focused on tumors in which the median expression of *NAGS*, *CPS1,* and citrin mRNA was higher than in the matched normal tissue. The median *NAGS*, *CPS1,* and citrin expression was between 1.4- and 4-fold higher in glioblastoma multiforme (GBM, n = 166), glioma (GBMLGG, n = 696), stomach adenocarcinoma (STAD, n = 415), and stomach and esophageal carcinoma (STES, n = 600) (Figure 1A,B). *NAGS* and *CPS1* had a similar median mRNA expression in lung adenocarcinoma (LUAD, n = 517) and matched normal tissue, while citrin was overexpressed in LUAD (Figure 1A,B).

*NAGS* and *CPS1* have no known functions in tissues other than liver and small intestine [10,11]. The citrin gene is expressed more broadly since it functions in ureagenesis and the malate-aspartate shuttle, but the liver has the highest citrin expression [24,25,26]. Therefore, we evaluated the *NAGS*, *CPS1,* and citrin mRNA and protein expression in glioblastoma multiforme, glioma, lung adenocarcinoma, stomach adenocarcinoma, and stomach and esophageal carcinoma in comparison to the expression of the three genes in the liver and small intestine. The GTEx portal was queried for *NAGS*, *CPS1,* and citrin mRNA expression levels. The abundance of the NAGS, CPS1, and citrin proteins in the liver, small intestine, esophagus, stomach, brain, and lung tissues was obtained from ProteomicsDB [39,40] and the GTEx project [41]. First, we compared the mRNA and protein expression of the three genes in the liver and small intestine to determine whether they are good proxies for NAGS, CPS1, and citrin protein expression. Differences in the abundance of *NAGS*, *CPS1,* and citrin mRNA and NAGS, CPS1, and citrin proteins between the liver and small intestine are similar (Figure 1C–K). This suggests that the abundances of *NAGS*, *CPS1,* and citrin mRNA are good proxies for the abundance of the NAGS, CPS1, and citrin proteins. *NAGS* mRNA is 1–2 orders of magnitude less abundant in the stomach, esophagus, brain, and lung tissues than in the liver (Figure 1C), while the NAGS protein was detectable only in the lung (Figure 1D,E). Abundance of the *CPS1* mRNA and protein is 2–3 orders of magnitude lower in the esophagus, stomach, brain, and lung tissues than in the liver (Figure 1F–H). This suggests markedly lower NAGS and CPS1 expression in glioblastoma multiforme, glioma, lung adenocarcinoma, stomach adenocarcinoma, and stomach and esophageal carcinoma than in the liver and small intestine. Citrin mRNA and protein expression are similar in the esophagus, stomach, brain, lung, and small intestine, and is approximately one tenth of the citrin expression levels in the liver (Figure 1I–K). Up to three-fold higher citrin expression in the tumors than in the matching normal tissue (Figure 1A,B) can affect the flux of metabolites in tumor cells. 

### 2.2. Expression of NAGS, CPS1, and Citrin in Tumors and Patient Outcomes

We queried the cBioPortal database [42,43] to determine whether the expression of *NAGS*, *CPS1,* and citrin in glioblastoma multiforme, glioma, stomach adenocarcinoma, and lung adenocarcinoma correlated with the patient outcomes for these four tumor types. Patient outcome data for stomach and esophageal carcinoma are not available in the cBioPortal; cBioPortal does have the outcome data for patients with glioblastoma, which is a type of glioma. Therefore, we queried cBioPortal for the association between *NAGS*, *CPS1*, and citrin mRNA expression levels and outcomes of patients with glioblastoma, glioblastoma multiforme, lung adenocarcinoma, and stomach adenocarcinoma. Glioblastoma, glioblastoma multiforme, lung adenocarcinoma, and stomach adenocarcinoma samples were ranked according to expression level of *NAGS*, *CPS1,* or citrin mRNA and divided into quartiles based on the expression levels of the three genes, followed by comparison of survival times of patients with the highest and the lowest expression of the three genes (Figure 2). The highest NAGS mRNA expression was associated with a worse outcome, i.e., shorter survival time, in patients with glioblastoma (Figure 2A). There was no association between *CPS1* and citrin mRNA expression levels and the outcome of patients with glioblastoma (Figure 2B,C). There was a trend (*p* = 0.07) towards a worse outcome (shorter survival time) for patients with glioblastoma multiforme exhibiting the highest expression of *NAGS* mRNA, compared to glioblastoma multiforme patients exhibiting the lowest expression of *NAGS* mRNA (Figure 2D). Expression levels of *CPS1* and citrin mRNA were not associated with the outcome of patients with glioblastoma multiforme (Figure 2E,F). There was a trend towards a worse outcome (*p* = 0.07) in patients exhibiting the lowest quartile of *NAGS* mRNA expression in lung adenocarcinoma (Figure 2G), while the outcome was worse for patients with the highest *CPS1* mRNA expression in lung adenocarcinoma (Figure 2H). There was no association between citrin mRNA expression and outcome of patients with lung adenocarcinoma (Figure 2I), and there was no association between expression levels of *NAGS*, *CPS1*, or citrin mRNA and outcome of patients with stomach adenocarcinoma (Figure 2J–L).

### 2.3. Epigenetic Regulation of NAGS, CPS1, and Citrin in Lung Adenocarcinoma

Overexpression of *NAGS* and *CPS1* genes in the brain, lungs, and stomach tissues–where these two genes are not highly expressed–would require epigenetic changes that allow the transcription of the two genes. The citrin gene is expressed in most tissues due to its functioning in the malate-aspartate shuttle [26]. Therefore, the epigenetic regulation of citrin ought to be similar in the liver, brain, lung and stomach. We sought to compare chromatin accessibility to transcription factors, histone modifications, binding of CTCF (a component of the cohesin complex that regulates chromatin domain organization), and a POLR2A subunit of RNA polymerase II as indicators of molecular mechanisms that enable gene expression in the liver, where *NAGS*, *CPS1*, and citrin are highly expressed; brain, lung and stomach tissues; and cancer cell lines that model glioblastoma, glioblastoma multiforme, lung adenocarcinoma, and stomach adenocarcinoma. DNase-Seq and ATAC-Seq data, which indicate chromatin accessibility to transcription factors; ChIP-Seq data for CTCF; POLR2A; H3K4me3, which indicate active gene transcription; and H3K27ac, which indicates active enhancers, were available in the ENCODE database for the liver and lung tissues, and for the A549 lung adenocarcinoma cell line. 

The *NAGS* promoter, −3 kb enhancer, and a regulatory element in the first intron of the *NAGS* gene regulate its expression in the liver [44,45] and are among candidate *cis*-regulatory elements (cCRE) of the *NAGS* gene (Figure 3A–C, highlighted in green). DNase–Seq and ATAC-Seq data indicate that the *NAGS* promoter and −3 kb enhancer are accessible to transcription factor binding in the liver (Figure 3A), but not in the A549 cell line and lung tissue (Figure 3B,C). An ATAC-Seq peak located approx. 9.5 kb upstream of the *NAGS* transcription start site (TSS) coincides with CTCF binding in the liver, A549 cells, and lung, suggesting that this region may represent a boundary between the *NAGS* locus and the upstream *PYY* locus (Figure 3A–C). The −3 kb *NAGS* enhancer and intronic regulatory elements are associated with strong H3K27ac signals in the liver, indicating that these two regulatory elements are active enhancers in the liver (Figure 3A). The H3K27ac signal is absent at the −3 kb enhancer in A549 cells and lung tissue (Figure 3B,C), consistent with the liver-specific activity of this regulatory element [45]. The H3K27ac signal at the *NAGS* intronic regulatory element is markedly weaker in A549 cells and lung tissue, indicating low *NAGS* expression (Figure 3B,C). The H3K4me3 and POLR2A signals were lower in A549 cells and lung tissue than in the liver (Figure 3A–C). Taken together, these data are consistent with low *NAGS* expression in the A549 cells, lung adenocarcinoma, and lung tissue. 

Expression of *CPS1* is regulated by a proximal enhancer and promoter located immediately upstream of *CPS1* TSS, and a distal enhancer located approx. 7.5 kb upstream of the *CPS1* TSS [45,46]. These known *CPS1* regulatory elements are associated with strong ATAC-Seq and DNase-Seq signals in the liver (Figure 3D and Appendix A, highlighted in green). Upstream of the *CPS1* distal enhancer are several cCREs, of which, some are associated with ATAC-Seq and DNase-Seq signals in the liver (Figure 3D and Appendix A). The distal *CPS1* enhancer is associated with strong ATAC-Seq and DNase-Seq signals in A549 cells (Figure 3E and Appendix A), but not in the lung (Figure 3F and Appendix A). The other known and predicted *CPS1* upstream regulatory elements are associated with weak ATAC-Seq and DNase-Seq signals in the lung and A549 cells (Figure 3E,F and Appendix A). Approximately 35 kb upstream of the *CPS1* TSS is a site that binds CTCF in the liver and is predicted to be a chromatin insulator (Figure 3D and Appendix A). This predicted chromatin insulator does not bind CTCF in the lung and A549 cells (Figure 3E,F). Known *CPS1* regulatory elements and some of the predicted *CPS1* regulatory elements are associated with strong H3K27ac, H3K4me3, and POLR2A signals in the liver (Figure 3D), which are weak in A549 cells (Figure 3E) and absent in the lung tissue (Figure 3F). This is consistent with aberrant *CPS1* expression in lung adenocarcinoma. 

Expression of the citrin gene is regulated by a recently characterized promoter [47,48]. This promoter is associated with strong ATAC-Seq, DNase-Seq, H3K27ac, H3K4me3, and POR2A signals in the liver, A549 cells, and lung tissue, consistent with ubiquitous citrin gene expression (Figure 3G–I and Appendix A). ATAC-Seq signals were associated with some of the cCREs in the first intron of the citrin gene and its upstream regulatory region (Figure 3G–I and Appendix A). CTCF binding was not detected in the first intron of the citrin gene and within 25 kb upstream of the citrin TSS (Figure 3G–I). However, CTCF binding sites were present approx. 100 kb upstream of the citrin TSS and within intron 4 of the citrin gene (Appendix A). Similar epigenetic regulation of citrin in the liver and lung is consistent with its expression in many tissues, and could contribute to its aberrant expression in lung adenocarcinoma.

**Figure 3 ijms-24-06754-f003:**
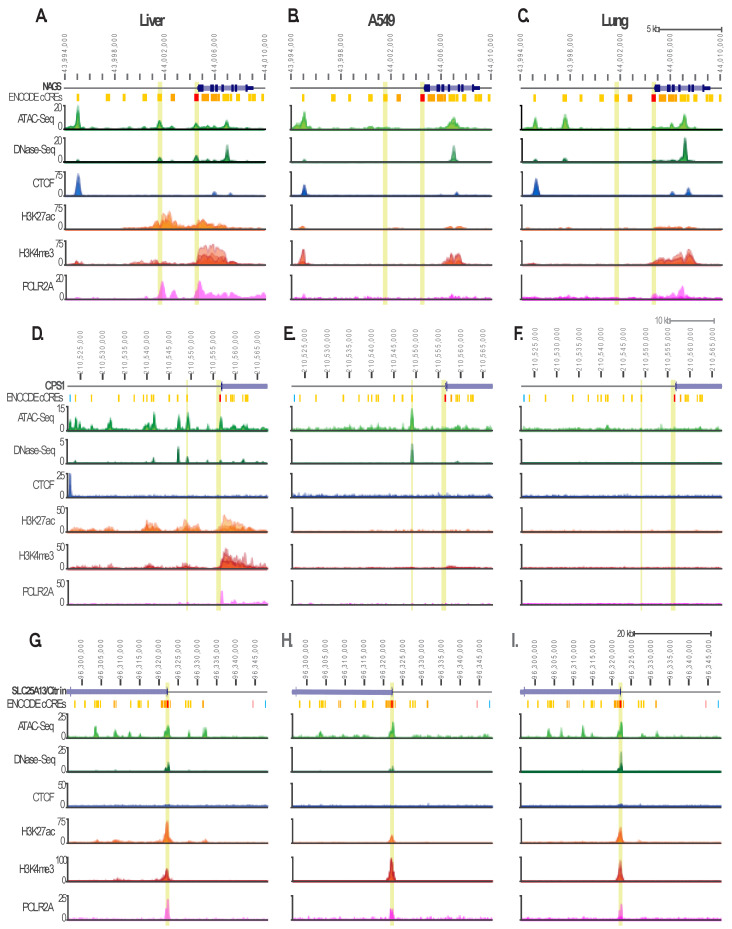
Epigenetic regulation of *NAGS* (**A**–**C**), *CPS1* (**D**–**F**), and citrin (**G**–**I**) in the liver tissue (**A**,**D**,**G**), A549 cell line (**B**,**E**,**H**), and lung tissue (**C**,**F**,**I**). Chromatin accessibility (ATAC-Seq and DNase-Seq shown in light and dark green, respectively), CTCF binding (blue), acetylation of histone H3 lysine 27 (H3K27ac, shown in light orange), tri-methylation of histone H3 lysine 4 (H3K4me3, shown in dark orange), and binding of the 2A subunit of RNA polymerase II (POLR2A, shown in magenta) are shown for the following genomic regions: whole *NAGS* gene, approx. 10.5 kb of the upstream regulatory region and 1 kb downstream of the *NAGS* (chr17: 43,994,000–44,010,000); first exon of the *CPS1* gene, approx. 35 kb of the upstream regulatory region and approx. 10 kb of the first *CPS1* intron (chr2: 210,522,000–210,567,000); first and second exons of the citrin gene, first intron of the citrin gene, and approx. 25 kb of the upstream regulatory region (chr7: 96,296,000–96,348,000). Predicted cCREs: promoters–red, proximal enhancers–orange, distal enhancers–yellow, chromatin insulators–blue.

### 2.4. Sequence Variation and Expression in Tumor and Normal Tissues

Since *NAGS*, *CPS1,* or citrin genes are not highly expressed in the brain, lung, and stomach, we explored molecular mechanisms that can contribute to elevated NAGS, CPS1, and citrin activities in tumors. Increased copy numbers of *NAGS*, *CPS1*, and citrin genes and/or activating sequence variants could result in higher activity of the three proteins in glioblastoma, glioblastoma multiforme, lung adenocarcinoma, and stomach adenocarcinoma.

#### 2.4.1. Copy Number Variation of *NAGS*, *CPS1*, and Citrin Genes in Glioblastoma, Glioblastoma Multiforme, Lung Adenocarcinoma and Stomach Adenocarcinoma

Structural variants resulting in increased copy numbers of *NAGS*, *CPS1*, and/or citrin loci could explain the overexpression of the three genes. Therefore, we queried cBioPortal for the number of glioblastoma, glioblastoma multiforme, lung adenocarcinoma, and stomach adenocarcinoma samples exhibiting *NAGS*, *CPS1*, and citrin gene copy number variation (CNV), and whether there is an association between CNVs and mRNA expression of the three genes in the four tumor types (Figure 4). CNVs are classified as deep deletions, shallow deletions, diploid, gain, and amplifications in cBopPortal, and they correspond to 0, 1, 2, a few additional copies, and more than a few additional copies of the gene/region of interest, respectively [49].

Glioblastoma and glioblastoma multiforme had similar patterns of *NAGS*, *CPS1,* and citrin CNV (Figure 4). *NAGS* and *CPS1* loci were diploid in the 82 and 89%, respectively, of glioblastoma samples while 5–10% of glioblastoma samples exhibited either shallow deletions or gains of NAGS and CPS1 genes (Figure 4A). Most of the glioblastoma samples (80%) exhibited gains of the citrin gene (Figure 4A). The citrin gene was diploid in 18% of glioblastoma samples (Figure 4A). The remaining glioblastoma samples had amplifications of the citrin gene (Figure 4A). There was no association between CNVs and the expression of *NAGS* and *CPS1* mRNA in glioblastoma (Figure 4B,C). The expression of citrin mRNA was higher in glioblastoma samples with a higher copy number of the citrin gene (Figure 4D). The majority of glioblastoma multiforme samples were diploid for *NAGS* and *CPS1* loci and exhibited gains of the citrin gene (Figure 4E). The remaining glioblastoma multiforme samples exhibited either shallow deletions or gains of the *NAGS* and *CPS1* genes, and either gains or amplifications of the citrin gene (Figure 4E). There was no association between CNVs and the expression of *NAGS* and *CPS1* mRNA in glioblastoma multiforme samples (Figure 4F,G). The expression of citrin mRNA was higher in the glioblastoma multiforme samples with a higher copy number of the citrin gene (Figure 4H).

More than 90% of the lung adenocarcinoma samples were diploid for *NAGS*, *CPS1,* and citrin genes (Figure 4I). The remaining lung adenocarcinoma samples exhibited deep and shallow deletions, gains, and amplifications of the three genes (Figure 4I–L). The expression of *NAGS*, *CPS1*, and citrin mRNA correlated with the CNV of each gene (Figure 4J–L). 

**Figure 4 ijms-24-06754-f004:**
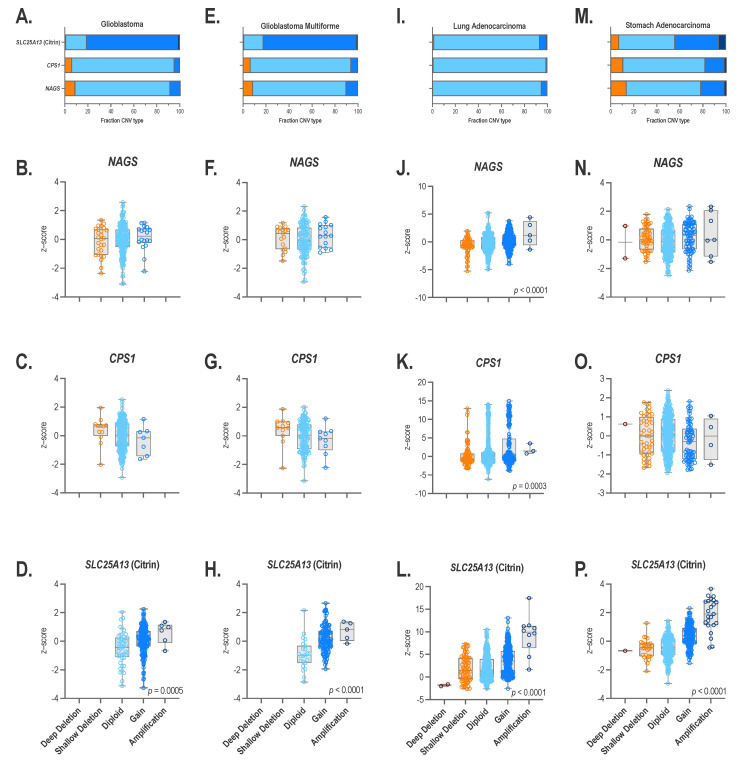
Copy number variation and *NAGS*, *CPS1,* and citrin mRNA expression. Fraction of glioblastoma (**A**), glioblastoma multiforme (**E**), lung adenocarcinoma (**I**) and stomach adenocarcinoma (**M**) samples with deep deletions, shallow deletions, two copies, gain and amplification of *NAGS*, *CPS1,* and citrin genes. Relationship between copy number and *NAGS* (**B**,**F**,**J**,**N**), *CPS1* (**C**,**G**,**K**,**O**) and citrin (**D**,**H**,**L**,**P**) mRNA expression. Brown–deep deletion, orange–shallow deletion, light blue–diploid, medium blue–gain, dark blue–amplification of the genomic regions of interest.

The CNV of *NAGS* and *CPS1* were similar in stomach adenocarcinoma samples. The majority of the stomach adenocarcinoma samples, 64% and 71%, were diploid for *NAGS* and *CPS1* genes, respectively; 20% and 16% of the stomach adenocarcinoma samples exhibited gains of *NAGS* and *CPS1*, respectively; 13% and 10% of stomach adenocarcinoma samples exhibited shallow deletions of *NAGS* and *CPS1*, respectively; and the remaining stomach adenocarcinoma samples had either deep deletions or amplifications of the two genes (Figure 4M). The majority of the stomach adenocarcinoma samples were either diploid (48%) or exhibited a gain (38%) of the citrin gene; the remaining samples exhibited shallow deletions, amplifications, and a loss of the citrin gene (Figure 4M). There was no association between the CNV and *NAGS* mRNA expression in the stomach adenocarcinoma samples (Figure 4N). The expression of *CPS1* mRNA was lower in the stomach adenocarcinoma samples with higher copy numbers of the *CPS1* gene (Figure 4O). The expression of citrin mRNA was higher in the stomach adenocarcinoma samples with higher copy number of the citrin gene (Figure 4P).

#### 2.4.2. Germline and Somatic *NAGS*, *CPS1*, and Citrin Sequence Variants

Metabolic dysregulation contributing to increased de novo pyrimidine biosynthesis could result from sequence variants that increase the activity of NAGS, CPS1, and citrin without the need for the overexpression of the corresponding genes. Therefore, we intended to compare functional effects of *NAGS*, *CPS1*, and citrin somatic sequence variants found in glioblastoma, glioblastoma multiforme, lung adenocarcinoma, and stomach adenocarcinoma with functional effects of germline variants found in patients with respective urea cycle disorders, and in individuals without known rare diseases. However, the fraction of glioblastoma, glioblastoma multiforme, lung adenocarcinoma, and stomach adenocarcinoma samples with sequence variants in *NAGS*, *CPS1*, and citrin was low (Table 1). Therefore, we compared the types of *NAGS*, *CPS1*, and citrin sequence variants found in all tumor samples with sequence variants found in individuals without known rare diseases, and patients with *NAGS*, *CPS1*, or citrin deficiencies to increase the power of comparison. 

*NAGS*, *CPS1,* and citrin sequence variants found in tumor samples were collected from the TCGA [50], COSMIC [51], and cBioPortal [42,43] databases of tumor genomic information. *NAGS*, *CPS1*, and citrin sequence variants present in individuals without rare genetic disorders were collected from the gnomAD database [52]. Pathogenic, likely pathogenic sequence variants, variants of uncertain significance, and variants with conflicting interpretation found in patients with *NAGS*, *CPS1*, or citrin deficiency were collected from ClinVar and LOVD [53] databases as well as from published case reports (Figure 5A). For all three genes, there was little overlap between variants found in tumors, patients, and gnomAD (Figure 5B). This is not surprising since mechanisms of mutagenesis differ in germline and tumor cells that often exhibit aberrant mismatch repair and proofreading [54]. 

Although there was little overlap between sequence variants found in patients with *NAGS*, *CPS1*, or citrin deficiencies, tumors and gnomAD, the frequencies of different types of sequence variants were similar in the three groups of samples (Table 2 and Figure 5C). Variants were classified based on their location in genes (regulatory, 5′-UTR, intron, and 3′-UTR) and based on their effect on the coding region and splicing (missense, nonsense, frameshift, synonymous, splicing variants of canonical splice sites, and splice region variants that affect base pairs flanking canonical splice sites).

Loss-of-function (LOF) variants that include nonsense, frameshift, and splicing variants were more frequent in patients with *NAGS*, *CPS1,* or citrin deficiency than in tumors and gnomAD (Figure 5C and Table 2). A higher frequency of LOF variants in patients with *NAGS*, *CPS1*, or citrin deficiency than in tumors and gnomAD is expected, since carriers of LOF variants in any of the three genes have normal ureagenesis and phenotype [10]. Very small fractions of sequence variants were found in the 5′-UTRs, splice regions and 3′-UTRs of all three genes (Figure 5C and Table 2). This is not surprising since these regions are minor portions of *NAGS*, *CPS1*, and citrin genes. Sequence variants found in *NAGS*, *CPS1*, and citrin introns were common in tumors and gnomAD (Figure 5C and Table 2). This is likely due to the whole genome sequencing of DNA from tumor samples and gnomAD, while sequencing of patient DNA focuses on the coding regions and canonical splice sites where most disease-causing sequence variants are found. Both absolute and relative numbers of sequence variants found in *CPS1* and citrin introns were higher than in *NAGS* introns (Figure 5C and Table 2), likely due to the small size of *NAGS* introns compared to *CPS1* and citrin introns. Synonymous *NAGS* and *CPS1* sequence variants were more frequent in tumor samples and gnomAD than in patients with *NAGS* or *CPS1* deficiency, while citrin synonymous sequence variants were less frequent in gnomAD than in patients and tumor samples (Figure 5C and Table 2). Surprisingly, whole genome sequencing of DNA from tumor samples and gnomAD did not reveal many variants in the upstream regulatory regions (Figure 5C and Table 2). This could be due to the poor annotation of gene regulatory regions in the current assembly of the human genome. Missense variants found in all three genes were either the largest or the second largest fraction of all sequence variants in patients, tumors, and gnomAD (Figure 5C and Table 2). Similar fractions of *NAGS* and citrin variants found in patients and tumor samples were missense variants, while they represented a slightly lower fraction of *NAGS* and citrin variants in gnomAD (Figure 5C and Table 2). This pattern was different for *CPS1* missense variants. Similar fractions of *CPS1* variants found in tumor samples and gnomAD were missense variants, while they represented the highest fraction of *CPS1* variants found in patients with *CPS1* deficiency (Figure 5C and Table 2). 

There was more overlap between missense variants found in patients and gnomAD than between somatic missense variants found in tumors and germline variants found in patients and gnomAD (Figure 6A). The REVEL functional effect predictor was used to evaluate the effects of missense variants on protein function. Missense variants with a REVEL score above 0.5 are considered damaging while missense variants with a REVEL score below 0.5 are considered tolerated [55]. The median REVEL scores of *NAGS*, *CPS1*, and citrin missense variants found in patients were all above 0.5 (Figure 6B–D). This is consistent with the reduced or absent enzymatic activity and/or decreased stability of mutant NAGS, CPS1, and citrin proteins found in patients with respective deficiencies. 

The median REVEL scores of NAGS missense variants found in tumors and gnomAD are below 0.5, suggesting that most of these variants do not affect NAGS protein function (Figure 6B). REVEL scores of CPS1 and citrin missense variants found in tumors and gnomAD are above 0.5 but lower than the median REVEL score in patients (Figure 6C,D). This likely reflects a higher conservation of CPS1 and citrin proteins across phyla [56] and the weight given to amino acid conservation by REVEL [55]. 

The strength of evidence for damaging and tolerated/benign effects of a missense variant on protein function can be inferred from its REVEL score [57]. The distribution of *NAGS* REVEL scores differs in patients vs. gnomAD and tumor samples. REVEL scores for the majority of *NAGS* missense variants in all three groups of samples indicate an uncertain effect on NAGS function. The second highest fraction (29%) of *NAGS* missense variants have REVEL scores indicative of moderate support for the damaging effect on NAGS function (Figure 6E).

The majority of *CPS1* missense variants found in patients with *CPS1* deficiency (40%) have REVEL scores higher than 0.932, suggesting strong evidence for damaging effects on CPS1 function (Figure 6F). Between 25 and 30% of sequence variants in all three groups of samples have REVEL scores that indicate moderate evidence for damaging effects on CPS1 function (Figure 6F). The majority of *CPS1* missense variants in gnomAD and tumor samples have REVEL scores that indicate an uncertain effect on CPS1 function (Figure 6F). 

Two-thirds of citrin missense variants found in patients with a citrin deficiency have REVEL scores that suggest either a moderate or uncertain effect on citrin function (Figure 6G). The majority of citrin missense variants in gnomAD and tumor samples have REVEL scores that indicate an uncertain effect on citrin function (Figure 6F). None of the missense variants found in *NAGS*, *CPS1,* and citrin genes had REVEL scores that indicate either strong or very strong evidence for a benign effect on protein function.

**Figure 6 ijms-24-06754-f006:**
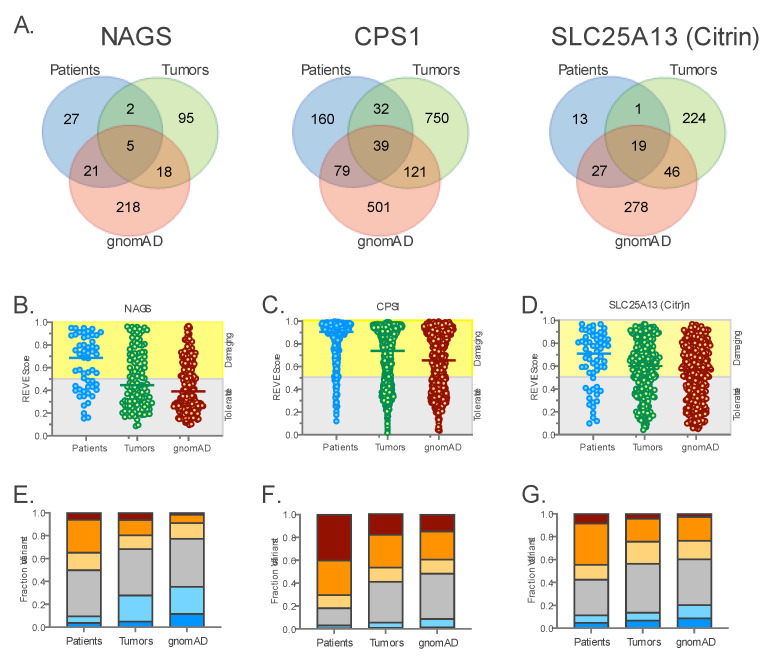
*NAGS*, *CPS1*, and citrin missense variants in urea cycle patients, tumors, and individuals without rare genetic disorders. (**A**) Venn diagrams showing overlap between *NAGS*, *CPS1,* or citrin missense variants found in patients with respective deficiencies, tumors, and gnomAD. (**B**–**D**) REVEL scores of *NAGS* (**B**), *CPS1* (**C**) and citrin (**D**) missense variants found in patients (blue), tumors (green), and gnomAD (orange). Yellow–REVEL scores that indicate damaging effect of missense variants on protein function; gray–REVEL scores indicating that missense variants are tolerated and do not affect protein function. (**E**–**G**) Fraction of sequence variants with REVEL scores indicating strength of evidence for either damaging or benign effect of missense variant on *NAGS* (**E**), *CPS1* (**F**) and citrin (**G**) function in patients with respective deficiencies, tumor samples, and gnomAD. Brown–strong evidence for damaging effect on protein function; orange–moderate evidence for damaging effect on protein function; light orange–supporting evidence for damaging effect on protein function; gray–uncertain effect on protein function; light blue–supporting evidence for benign effect on protein function; dark blue–moderate evidence for benign effect on protein function.

### 2.5. NAGS, CPS1, and Citrin Expression in Individual Tumor Samples

NAG, produced by NAGS, is required for CPS1 function in vivo [11,12,13,14,15,16,17,18,58,59]. Increased production of carbamylphosphate (CP) due to the overexpression of CPS1, together with higher cytoplasmic abundance of aspartate due to the overexpression of citrin could contribute to a higher CAD activity and de novo pyrimidine synthesis. Therefore, we examined whether individual glioblastoma, glioblastoma multiforme, lung adenocarcinoma, and stomach adenocarcinoma samples exhibit the overexpression of *NAGS*, *CPS1*, and citrin mRNA. The correlation between *NAGS* and *CPS1*, citrin and *NAGS*, and *CPS1* and citrin mRNA expression in individual tumor samples was either weak or negligible (Figure 7). CPS1 and citrin protein abundance data were available for individual glioblastoma and lung adenocarcinoma samples [60,61]. The correlation between CPS1 and citrin protein abundance was weak in individual glioblastoma and lung adenocarcinoma samples (Figure 8). These results suggest that an increased cytoplasmic supply of either CP or aspartate may be sufficient for a higher de novo pyrimidine biosynthesis, as well as the need for an alternative explanation of CPS1 activity in the absence of *NAGS* expression and NAG. 

## 3. Discussion

This study used data mining to assess whether the aberrant expression of *NAGS* and biosynthesis of NAG can explain the increased activity of CPS1 that contributes to higher de novo pyrimidine production and tumor cell proliferation. We also assessed whether tumors that overexpress *NAGS* and *CPS1* exhibit a higher expression of citrin, which can increase the cytoplasmic supply of aspartate for the de novo biosynthesis of pyrimidine nucleotides. The project was conceived and carried out during the COVID-19 pandemic when access to laboratory research was limited. We asked a series of questions that could be answered using data mining and bioinformatic approaches and used the answers to generate experimentally testable hypotheses. 

The urea cycle can catalyze the formation of approximately 20 g of urea nitrogen, which corresponds to 43 g of urea per day in an average man [62]. Urea cycle enzymes and transporters are among the most abundant proteins in periportal hepatocytes [63]. The abundances of *NAGS*, *CPS1,* and citrin mRNA and proteins in the lung, brain, esophagus, and stomach tissues are 10–1000 times lower than in the liver, and the expression of the three genes is two- to four-fold higher in glioblastoma, glioblastoma multiforme, lung adenocarcinoma, stomach adenocarcinoma, and stomach and esophagus carcinoma samples. Although markedly lower than in the liver, the expression of *CPS1* in lung cancers and p53–depleted tumor cell lines can sufficiently alter cellular homeostasis towards the increased de novo biosynthesis of pyrimidine nucleotides via a higher CAD activity [31,32,34]. The NAGS protein is expressed in two cell lines commonly used to model non-small cell lung carcinoma [36]; this provides a molecular mechanism for CPS1 activity and increased production of CP in the two cell lines. Since CPS1 is a mitochondrial enzyme, CP produced by CPS1 in the mitochondria of tumor cells would have to be translocated into the cytoplasm to enter de novo pyrimidine biosynthesis. Excess CP, produced in hepatocytes of patients with OTC deficiency, can exit mitochondria and enter pyrimidine biosynthesis leading to the accumulation of orotic acid, which is an intermediate of pyrimidine biosynthesis and a biomarker of OTC deficiency [10,11]. 

ARALAR1 is a citrin paralog with identical biochemical function [22]. ARALAR1 is highly expressed in the brain, skeletal muscle and heart, present in other tissues and absent from the liver, the site of high citrin expression [22]. An increased supply of cytoplasmic aspartate due to the ectopic overexpression of citrin could disrupt the balance of cytoplasmic metabolites, resulting in a higher CAD activity, higher de novo pyrimidine biosynthesis, and metabolic reprogramming that leads to increased cell proliferation. In addition to supplying cytoplasmic aspartate for increased CAD activity, the overexpression of citrin in tumor cells could lead to metabolic reprogramming in tumor cells through the disruption of the malate-aspartate shuttle and energy production [29,30]. Therefore, we hypothesize that the overexpression of NAGS, CPS1, and citrin promotes the proliferation of glioblastoma, glioblastoma multiforme, stomach adenocarcinoma, and stomach and esophagus carcinoma cells through the increased biosynthesis of pyrimidine nucleotides and/or dysregulated malate-aspartate shuttle. This hypothesis could be tested in glioblastoma, stomach, and esophageal cancer cell lines by determining whether they express *NAGS*, *CPS1,* and/or citrin genes and proteins, followed by measuring the rate of cell proliferation and metabolite concentrations after knocking down the expression of the three genes. 

Copy number variation, aberrant epigenetic regulation, and gain-of-function sequence variants are molecular mechanisms that could contribute to the high activity of NAGS, CPS1, and citrin in tumor cells that originate from tissues that normally do not express the three genes. An increased gene copy number appears to be responsible for citrin overexpression in glioblastoma, glioblastoma multiforme, lung adenocarcinoma, and stomach adenocarcinoma samples. Unlike the four tumor types examined here, amplification of the chromosomal region that harbors the citrin locus has been observed in hepatocellular carcinoma samples, but the abundance of citrin mRNA was similar in the tumors and neighboring normal tissue [64]. 

An altered chromatin structure of *NAGS*, *CPS1*, and citrin loci in tumors originating from the brain, lung, stomach, or esophagus tissues would allow the binding of transcription factors and RNA polymerase II to promoters and enhancers of the three genes. Histone modifications, CTCF, and RNA polymerase II binding to *NAGS*, *CPS1,* and citrin regulatory regions in the liver, lung, and A549 lung adenocarcinoma cells suggest that all three genes appear to be poised for expression in the lung tissue. Reporter gene assays have shown that the *NAGS* promoter functions in the A549 cell line, but the −3 kb *NAGS* enhancer does not [45]. This could explain the low level of *NAGS* expression in the lung. Therefore, we hypothesize that *CPS1* and citrin promoters could function in the glioblastoma, glioblastoma multiforme, stomach adenocarcinoma, and stomach and esophagus carcinoma cells. This hypothesis could be tested with expression constructs in which *NAGS*, *CPS1*, or citrin promoters and enhancers control reporter gene expression in cell lines that model glioblastoma, glioblastoma multiforme, stomach adenocarcinoma, and stomach and esophagus carcinoma. 

There was little overlap between somatic sequence variants found in tumor samples and germline sequence variants found in patients with *NAGS*, *CPS1,* and citrin deficiencies, and in individuals without rare genetic diseases. This is not surprising, because defective DNA proofreading and mismatch repair as well as the imbalance of purine and pyrimidine pools in tumor cells results in different types of nucleotide replacements [54]. Gain-of-function sequence variants in gene regulatory elements can cause the increased expression and/or activity of NAGS, CPS1, and citrin in tumors. The data in the databases of somatic sequence variants was insufficient to evaluate whether sequence variants in *NAGS*, *CPS1,* and citrin promoters and enhancers contribute to the overexpression of the three genes in tumors. *NAGS*, *CPS1*, and citrin missense variants found in tumors are predicted to be less likely to damage protein function than missense variants found in patients with respective gene deficiencies. This could reflect the limitation of computational methods to predict gain-of-function effects of amino acid replacements. Gain-of-function missense variants that result in a higher enzymatic activity of mutant proteins have been observed in urea cycle enzymes. A high-throughput functional assay of all single nucleotide variant (SNV)-accessible amino acid replacements in human OTC revealed that around 5% of all OTC variants have up to 30% higher activity than wild-type human OTC [65]. Furthermore, human OTC with two common p.R46K and p.Q270R sequence variants had 40% higher enzymatic activity than the wild-type or either of the single mutant proteins [66]. Therefore, we hypothesize that some of the NAGS, CPS1, and citrin missense variants found in tumors can increase the activity of mutant proteins. This hypothesis could be tested in a high-throughput functional screen in *S. cerevisiae,* since this organism has homologs of NAGS, CPS1, and citrin [67,68]. 

Our analysis did not show an association between aberrant citrin expression and unfavorable outcomes in patients with glioblastoma, glioblastoma multiforme, and lung adenocarcinoma. There was no association between the expression of *NAGS*, *CPS1*, or citrin and stomach adenocarcinoma patient outcomes. The number of tumor samples with aberrant expression of two out of the three genes analyzed in this study was insufficient for evaluation of patient outcomes. Association between *NAGS* overexpression and unfavorable outcomes in patients with glioblastoma and glioblastoma multiforme suggests that the aberrant overexpression of NAGS and production of NAG could promote tumor development by activating CPS1 without the requirement for a correlated expression of *NAGS* and *CPS1* in tumor cells. Aberrant overexpression of NAGS could contribute to poor outcomes for patients with glioblastoma and glioblastoma multiforme independently of CPS1. NAG has been detected in the cytoplasm of mammalian and avian brain cells [69,70] but the function of NAG in the brain remains poorly understood. The abundance of NAG in the mammalian brain and liver are similar [69]. NAG that is produced in the liver mitochondria is transported to the cytoplasm for degradation [71]. A similar transport mechanism may operate in the brain cells allowing NAG, produced in the mitochondria of brain cells by the aberrantly overexpressed NAGS, to be transported in the cytoplasm, increase the pool of cytoplasmic NAG, and contribute to metabolic reprogramming and tumorigenesis. 

Our analysis replicated the association between *CPS1* overexpression and unfavorable outcomes in patients with lung adenocarcinoma [31]. However, the association between lower *NAGS* expression and unfavorable outcomes in patients with lung adenocarcinoma is consistent with the absence of correlated *NAGS* and *CPS1* expression in this tumor type. These observations were supported by the very weak correlation between *NAGS*, *CPS1*, and citrin expression in the individual glioblastoma, glioblastoma multiforme, lung adenocarcinoma, and stomach adenocarcinoma samples. This raises the question regarding the mechanism behind aberrant CPS1 activity that contributes to metabolic reprogramming towards increased de novo pyrimidine biosynthesis in tumor cells. One possibility is that CPS1 missense variants found in tumors enable CPS1 enzymatic activity without NAG. This hypothesis could be tested in a high-throughput functional activity screen in yeast cells that do not produce NAG. Another possibility is the activation of CPS1 in tumor cells by N-carbamylglutamate (NCG) produced by the human microbiota. NCG is a structural analog of NAG that can cross plasma and mitochondrial membranes to bind and activate CPS1 in patients with a NAGS deficiency [59]. NCG is an intermediate of histidine catabolism in bacteria and has been detected in metabolomes from gut and oral bacteria [72,73,74]. Therefore, it is possible that NCG of microbial origin can activate CPS1 in tumor cells. 

The connection between increased de novo biosynthesis of pyrimidine nucleotides, tumor cell proliferation, and aberrant CPS1 expression and activity made CPS1 an attractive target for antitumor drugs. AT067-H09 and H3B-120 are two inhibitors of CPS1 in vitro, in cultured hepatocytes and in cultured non-small cell lung carcinoma cell lines that appear to be promising antitumor drug candidates [36,75]. In addition to inhibiting aberrantly expressed CPS1 in tumor cells, AT067-H09 and H3B-120 could inhibit hepatic CPS1 and block the urea cycle leading to hyperammonemia, which can cause irreversible brain injury [10]. Therefore, preclinical and clinical testing of AT067-H09 and H3B-120 will require close collaboration between oncologists and physicians who treat patients with urea cycle disorders, as well as patients with acute and chronic liver failure to avoid hyperammonemia as a serious side effect of tumor treatment with drugs such as AT067-H09 and H3B-120. 

## 4. Methods and Materials

### 4.1. Expression Patterns of Urea Cycle Genes

The Firehose Data portal, developed by the Genome Data Analysis Center (GDAC) at the Broad Institute, was used to query gene expression profiles of tumors and matched normal tissues for the expression levels of all six urea cycle enzymes and two transporters. Median Log_2_RSEM values [37,38] for tumor samples and the matched normal tissues were extracted using the Firebrowse API (Appendix A). The median Log_2_RSEM values for urea cycle genes in tumor and matched normal tissue samples were determined by GDAC. The median Log_2_RSEM values from 28 tumor samples and matched normal tissues were used to calculate the fold-change values. For each urea cycle gene in each tumor type, the median Log_2_RSEM value for the matched normal tissue samples was subtracted from the median Log_2_RSEM value in tumor samples. The fold-change values were then calculated as two raised to the power of the difference between the median Log2RSEM values in tumor samples and matched normal tissues. 

The abundance of urea cycle gene transcripts in the liver, small intestine, stomach, esophageal mucosa and smooth muscle, cerebellum, cerebral cortex, and lung tissues were obtained from the GTEx Project database (Appendix A). The GTEx database (https://gtexportal.org/home/datasets) was accessed on 26 June 2022. The abundance of urea cycle enzymes and transporters in the liver, small intestine, stomach, esophagus, brain, and lung tissues were obtained from the GTEx Project [41] and Proteomics DB, which were accessed on 23 June 2022 [39,40] (Appendix A, respectively). 

The transcript abundance data for individual samples of lung adenocarcinoma [60,76,77,78,79,80], glioblastoma [61,79,81,82], glioblastoma multiforme [79,83], and stomach adenocarcinoma [50,79,83,84,85,86,87,88,89] were obtained from the cBioPortal for Cancer Genomics [42,43]. Transcript abundance was expressed either as RSEM in glioblastoma, glioblastoma multiforme, and stomach adenocarcinoma samples, or z-score relative to diploid cells in lung adenocarcinoma, to include the maximal number of samples for comparison (Appendix A). 

### 4.2. Patient Outcome Data

The correlation between patient outcomes and *NAGS*, *CPS1,* or citrin mRNA expression levels was available for select studies of glioblastoma [81], glioblastoma multiforme [79], lung adenocarcinoma [79,80], and stomach adenocarcinoma [79] at the cBioPortal. *NAGS*, *CPS1,* and citrin gene-specific charts were selected for each study. Gene-specific expression data for *NAGS*, *CPS1,* or citrin were used to define patient groups based on the quartiles of gene expression levels. Kaplan–Meier survival curves were compared for patients in the highest and the lowest expression quartile for each gene (Appendix A). 

### 4.3. Sequence Variant Data

CNV data for *NAGS*, *CPS1,* and citrin were available at the cBioPortal for select studies of glioblastoma [61,81], glioblastoma multiforme [50], lung adenocarcinoma [60,78,80], and stomach adenocarcinoma [50]. The number of samples harboring deep deletions, shallow deletions, two copies, gain and amplification of *NAGS*, *CPS1,* or citrin genes (Appendix A), as well as the correlation between the type of CNV and *NAGS*, *CPS1,* or citrin expression levels (Appendix A) were obtained for the four tumor types. 

The following databases and websites were queried in September 2021 to collect *NAGS*, *CPS1,* and citrin single nucleotide sequence variants and small indels found in (1) patients with *NAGS*, *CPS1,* or citrin deficiency; (2) tumor samples; and (3) individuals without rare genetic diseases: ClinVar (https://www.ncbi.nlm.nih.gov/clinvar/ (accessed on 1 May 2021)), Leiden Open Variant Database (LOVD) [53], COSMIC [51], cBioPortal [42,43], TCGA Data Portal [50], and gnomAD [52]. Published reports of sequence variants found in patients with *NAGS*, *CPS1,* and citrin deficiencies were also included in the analysis. *NAGS*, *CPS1*, and citrin sequence variants annotated as pathogenic, likely pathogenic, variants of unknown significance, or variants with conflicting interpretation in ClinVar and/or LOVD were included in the analysis. Variant validator [90] was used for the formatting of all sequences found in patients, tumors, and healthy individuals (Appendix A). Sequence variants were classified by type and location in the *NAGS*, *CPS1,* and citrin genes: regulatory (located upstream of transcription start site), 5′-UTR, missense, nonsense, frameshift, in-frame indel, synonymous, splicing (affecting canonical GT and AG splice signals at the 5′ and 3′-ends of introns), splice region (located up to 10 bp downstream and upstream of the GT and AG splice signals, respectively), intronic (located more than 10 bp away from the canonical splice sites), 3′-UTR, and unknown. Nonsense, frameshift, and splicing sequence variants were considered loss-of-function variants. The REVEL functional effect predictor was used to evaluate the effects of missense variants on protein function. REVEL [55] scores of *NAGS*, *CPS1*, and citrin missense variants obtained through the Ensembl Variant Effect Prediction tool [91] were used to evaluate the effects of missense variants on protein function (Appendix A). Missense variants with a REVEL score above 0.5 are considered damaging, while missense variants with a REVEL score below 0.5 are considered tolerated [55]. The ClinGen recommendations for relating the REVEL score of a missense variant to the strength of evidence that the missense variant has either a damaging or benign effect on protein function [57] were used to refine the functional effect predictions for *NAGS*, *CPS1*, and citrin variants found in patients, tumors, and gnomAD. A REVEL score greater than or equal to 0.932 is considered strong evidence for the damaging effect of a sequence variant on protein function [57]. A REVEL score between 0.932 and 0.773 is considered moderate evidence for the damaging effect of a sequence variant on protein function [57]. A REVEL score between 0.773 and 0.664 is considered supporting evidence for the damaging effect of a sequence variant on protein function [57]. The effect of a sequence variant with REVEL score between 0.644 and 0.290 on protein function is uncertain [57]. REVEL scores between 0.290 and 0.183, 0.193 and 0.016, and 0.016 and 0.003 are considered respectively as supporting, moderate, and strong evidence for benign effect on protein function, while a REVEL score lower than 0.003 is considered very strong evidence for benign effect on protein function [57]. 

### 4.4. Epigenetic Regulation Data

The ENCODE Project Functional Genomics Portal [92,93] was queried for the availability of data for DNase sensitivity and hypersensitivity sites, histone modifications, CTCF binding sites, and RNA polymerase II binding sites in the liver, lung, adult brain, and stomach tissues, as well as cancer cell lines that model glioblastoma, glioblastoma multiforme, lung adenocarcinoma, and stomach adenocarcinoma. The following filters were applied to the ENCODE Experimental Matrix: DNA binding and DNA accessibility for assay type; TF-ChIP-seq, Histone ChIP-seq, DNase-seq and ATAC-seq for assay title; Homo sapiens for organism; cell line and tissue for biosample classification; brain, liver, lung, stomach, A549, H54, A172 and M059J for biosample; liver, lung, brain, and stomach for organ. Histone ChIP-seq, TF-ChIP-seq, DNase-seq, and ATAC-seq data were available for liver, stomach, and lung tissues, and for the A549 lung adenocarcinoma cell line. Histone ChIP-seq, TF-ChIP-seq, and DNase-seq data were available for embryonic brain samples and brain samples from patients with Alzheimer’s disease. BigWig fold-change files for biological and technical replicates for the following experiments were downloaded: CTCF ChIp-seq, RNAP2A ChIP-seq, DNase-seq, ATAC-seq, H3K27Ac ChIP-seq, and H3K4me3 ChIP-seq. Genomic coordinates corresponding to *NAGS*, *CPS1,* and citrin loci were determined based either on the location of CTCF binding sites or the adjacent upstream and downstream gene. 

A custom Python script (https://github.com/MIMOR02/bigwig-file-vizualizations (accessed on 10 June 2022)) was used to extract and graph the fold-change signal from BigWig files for the *NAGS*, *CPS1*, and citrin loci. There are three notebooks in the GitHub repository which can be run in JupyterLab or Google Colab: Data_Visualization-GITHUB_PUSH.ipynb, ENCODE_JSON_parser-GITHUB_PUSH.ipynb, and Format_Directory-GITHUB_PUSH.ipynb. ENCODE_JSON_parser-GITHUB_PUSH.ipynb scrapes an experiment matrix from the ENCODE Project website for the desired bigwig file download links. Bigwig files were downloaded manually. Format_Directory-GITHUB_PUSH.ipynb creates a nested file system that all downloaded bigwig files will be sorted into. Data_Visualization-GITHUB_PUSH.ipynb reads in bigwig files, generates appropriate graphs to the user’s specifications, and moves the visualizations into the file system created by Format_Directory-GITHUB_PUSH.ipynb.

## Figures and Tables

**Figure 1 ijms-24-06754-f001:**
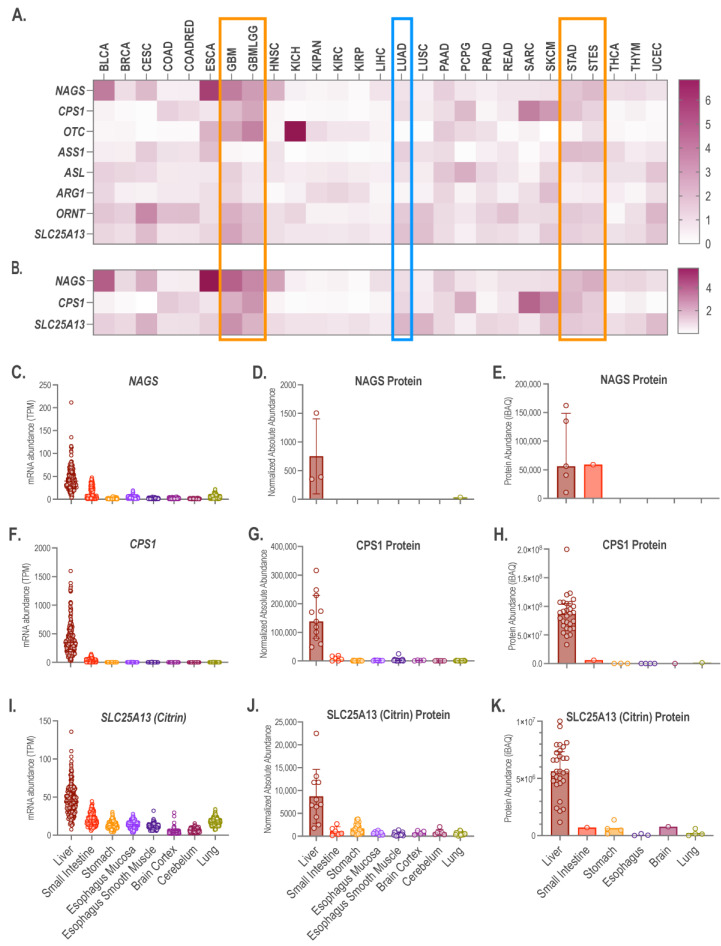
Expression of urea cycle genes in tumors and normal tissues. Fold-change of the median mRNA expression of all eight urea cycle genes (**A**) and NAGS, CPS1 and citrin mRNA (**B**) in tumor samples and the matched normal tissues. Depending on the tumor type, the number of tumor samples considered here was between 36 and 1100. Orange boxes—tumors that overexpress *NAGS*, *CPS1,* and citrin. Blue box—lung adenocarcinoma overexpressing citrin. Urea cycle gene symbols: *NAGS*–N-acetylglutamate synthase; *CPS1*–carbamylphosphate synthetase; *OTC*–ornithine transcarbamylase; *ASS1*–argininosuccinate synthetase 1; *ASL*–argininosuccinate lyase; *ARG1*–arginase 1; *ORNT*–ornithine transporter encoded by the *SLC25A15* gene; *SLC25A13*–glutamate/aspartate transporter, also known as citrin. Abundance of *NAGS* (**C**), *CPS1* (**F**) and citrin (**I**) mRNA in the liver, small intestine, stomach, esophagus, brain, and lung tissues in GTEx project. TPM–Transcripts per kilobase per million reads. Abundance of NAGS (**D**), CPS1 (**G**) and citrin (**J**) proteins in the liver, small intestine, stomach, esophagus, brain, and lung tissues in the GTEx project. Abundance of NAGS (**E**), CPS1 (**H**) and citrin (**K**) in the liver, small intestine, stomach, esophagus, brain, and lung tissues reported in ProteomicsDB. iBAQ–total intensity of precursor peptide ions divided by the number of theoretically observable peptides of the protein. Tumor type symbols: BLCA–bladder urothelial carcinoma; BRCA–breast invasive carcinoma; CESC–cervical and endocervical cancers; COAD–colon adenocarcinoma; COADRED–colorectal adenocarcinoma; GBM–glioblastoma multiforme; GBMLGG–glioma; HNSC–head and neck squamous cell carcinoma; KICH–kidney chromophobe; KIPAN–pan-kidney cohort (KICH + KIRC + KIRP); KIRC–kidney renal clear cell carcinoma; KIRP–kidney renal papillary cell carcinoma; LIHC–liver hepatocellular carcinoma; LUAD–lung adenocarcinoma; LUSC–lung small cell carcinoma; PPAD–pancreatic adenocarcinoma; PCPG–pheochromocytoma and paraganglioma; PRAD–prostate adenocarcinoma; READ–rectum adenocarcinoma; SARC–sarcoma; SKCM–skin cutaneous melanoma; STAD–stomach adenocarcinoma; STES–stomach and esophageal carcinoma; THCA–thyroid carcinoma; THYM–thymoma; UCEC–uterine corpus endometrial carcinoma.

**Figure 2 ijms-24-06754-f002:**
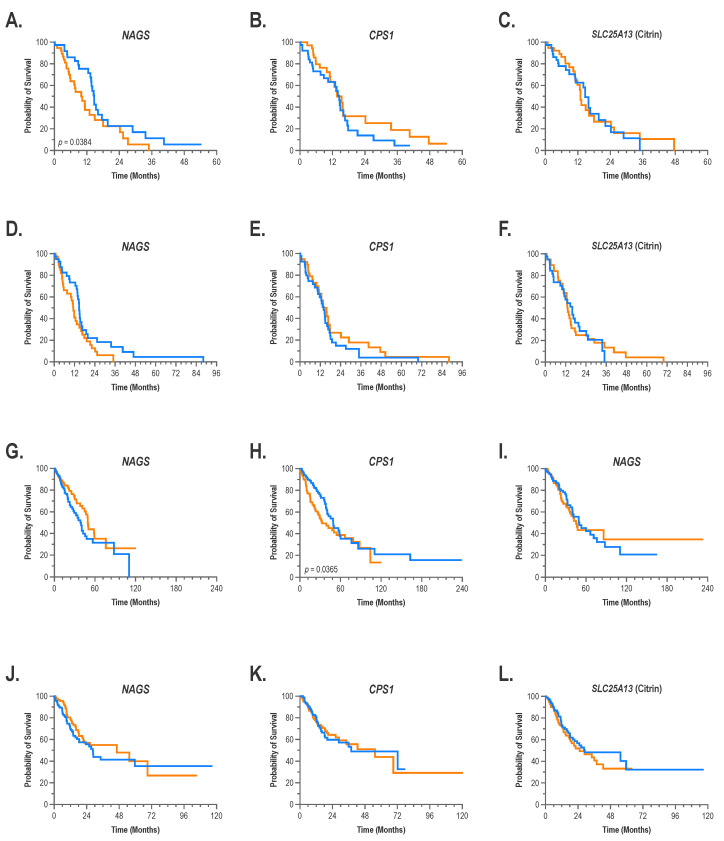
Association between expression of *NAGS*, *CPS1,* and citrin mRNA in tumors and patient outcomes. Kaplan–Meier curves showing survival of patients with glioblastoma (**A**–**C**), glioblastoma multiforme (**D**–**F**), lung adenocarcinoma (**G**–**I**) and stomach adenocarcinoma (**J**–**L**), exhibiting highest and lowest levels of *NAGS*, *CPS1*, and citrin mRNA expression. Orange–tumors in the highest quartile of mRNA expression; blue–tumors in the lowest quartile of mRNA expression.

**Figure 5 ijms-24-06754-f005:**
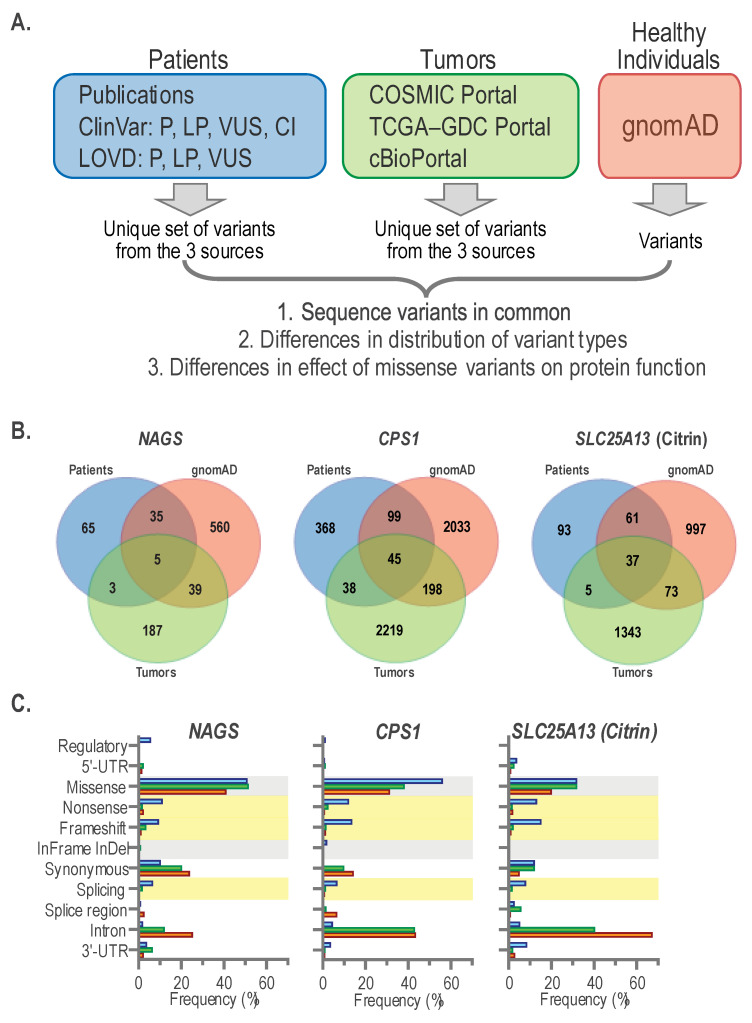
*NAGS*, *CPS1*, and citrin single nucleotide variants and small indels in urea cycle patients, tumors, and individuals without rare genetic diseases. (**A**) Strategy used to collect and analyze sequence variants found in the three groups of samples. (**B**) Venn diagrams showing overlap between *NAGS*, *CPS1*, or citrin sequence variants found in patients with respective deficiencies, tumors and gnomAD. (**C**) Distribution of *NAGS*, *CPS1,* and citrin variant types and their functional effects in patients (blue), tumors (green) and gnomAD (orange).

**Figure 7 ijms-24-06754-f007:**
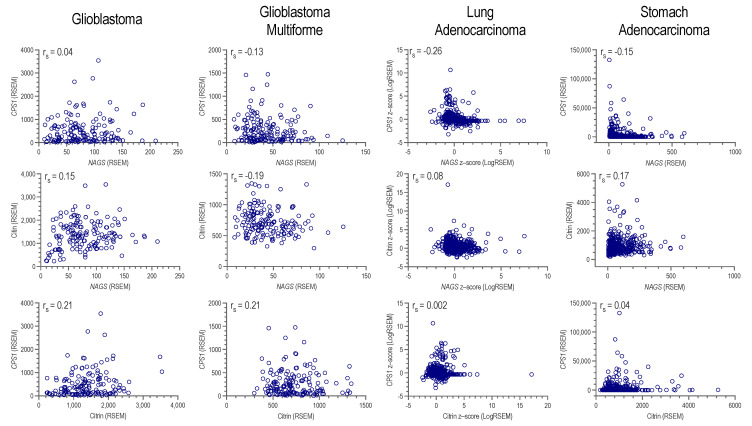
Correlation between *NAGS* and *CPS1* mRNA expression (**top**), *NAGS* and citrin mRNA expression (**middle**), and citrin and *CPS1* mRNA expression (**bottom**) in individual glioblastoma, glioblastoma multiforme, lung adenocarcinoma, and stomach glioblastoma samples. Spearman correlation coefficient was calculated to infer the relationship between mRNA expression levels in individual tumor samples.

**Figure 8 ijms-24-06754-f008:**
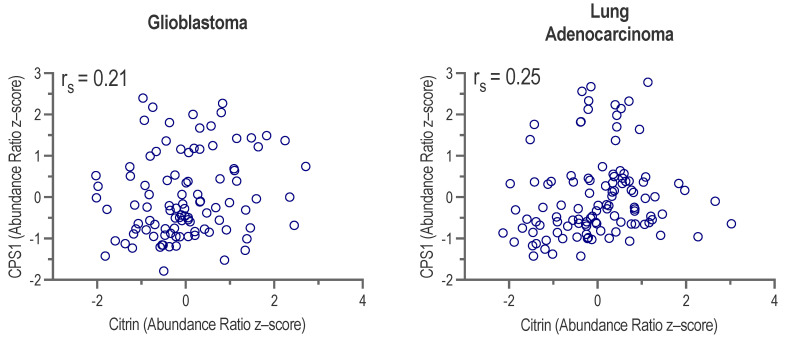
Correlation between citrin and CPS1 protein abundance in individual glioblastoma (**left**) and lung adenocarcinoma (**right**) samples. Spearman correlation coefficient was calculated to infer the relationship between protein abundance in individual tumor samples.

**Table 1 ijms-24-06754-t001:** Number and fraction of glioblastoma, glioblastoma multiforme (GBM), lung adenocarcinoma (LUAD), and stomach adenocarcinoma (STAD) samples with sequence variants in *NAGS*, *CPS1,* and citrin genes.

	Glioblastoma and GBM	LUAD	STAD
NAGS	7 ^a^/2176 ^b^ (0.32%) ^c^	11/1737 (0.63%)	6/1208 (0.50%)
CPS1	41/2660 (1.54%)	229/2313 (9.90%)	74/1261 (5.87%)
Citrin	42/1471 (2.86%)	44/2153 (2.04%)	31/1240 (2.50%)

^a^ Number of tumor samples with SNVs and short InDels in COSMIC and cBioPortal. ^b^ Total number of analyzed tumor samples in COSMIC and cBioPortal. ^c^ Percent of tumor samples with SNVs and Indels.

**Table 2 ijms-24-06754-t002:** Number and type of sequence variants found in *NAGS*, *CPS1,* and citrin genes in patients with *NAGS*, *CPS1,* or citrin deficiency, tumor samples, and individuals without rare genetic disorders.

** *NAGS* **	**Patients** **Count (%)**	**Tumors** **Count (%)**	**GnomAD** **Count (%)**
Regulatory	6 (5.6)	0 (0.0)	0 (0.0)
5’-UTR	0 (0.0)	5 (2.1)	9 (1.4)
Missense	55 (50.9)	120 (51.5)	262 (41.0)
Nonsense	12 (11.1)	4 (1.7)	14 (2.2)
Frameshift	10 (9.3)	8 (3.4)	7 (1.1)
InFrame InDel	0 (0.0)	2 (0.9)	1 (0.2)
Synonymous	11 (10.2)	47 (20.2)	152 (23.8)
Splicing	7 (6.5)	4 (1.7)	3 (0.5)
Splice region	1 (0.9)	0 (0.0)	16 (2.5)
Intron	2 (1.9)	28 (12.0)	161 (25.2)
3’-UTR	4 (3.7)	15 (6.4)	14 (2.2)
Total	108	233	639
** *CPS1* **	**Patients**	**Tumors**	**gnomAD**
Regulatory	6 (1.1)	0 (0.0)	0 (0.0)
5’-UTR	4 (0.7)	28 (1.1)	13 (0.5)
Missense	312 (56.0)	956 (38.2)	743 (31.3)
Nonsense	67 (12.0)	61 (2.4)	16 (0.7)
Frameshift	76 (13.6)	38 (1.5)	30 (1.3)
InFrame InDel	10 (1.8)	4 (0.2)	8 (0.3)
Synonymous	0 (0.0)	244 (9.8)	340 (14.3)
Splicing	37 (6.6)	33 (1.3)	17 (0.7)
Splice region	0 (0.0)	35 (1.4)	155 (6.5)
Intron	25 (4.5)	1076 (43.0)	1029 (43.4)
3’-UTR	20 (3.6)	25 (1.0)	22 (0.9)
Total	557	2500	2373
***SLC25A13* (Citrin)**	**Patients**	**Tumors**	**gnomAD**
Regulatory	0 (0.0)	0 (0.0)	0 (0.0)
5’-UTR	7 (3.7)	29 (2.5)	13 (0.9)
Missense	61 (31.9)	372 (31.9)	291 (20.0)
Nonsense	25 (13.1)	20 (1.7)	28 (1.9)
Frameshift	29 (15.2)	24 (2.1)	14 (1.0)
InFrame InDel	0 (0.0)	3 (0.3)	1 (0.1)
Synonymous	23 (12.0)	140 (12.0)	71 (4.9)
Splicing	15 (7.9)	20 (1.7)	8 (0.5)
Splice region	5 (2.6)	66 (5.7)	11 (0.8)
Intron	10 (5.2)	470 (40.3)	980 (67.2)
3’-UTR	16 (8.4)	22 (1.9)	41 (2.8)
Unknown	191	1166	1458

## Data Availability

All data used in this study have been included in Appendix A. Custom Python scripts used for acquiring and analysis of the data from the ENCODE Project are available at https://github.com/MIMOR02/bigwig-file-vizualizations (accessed on 10 June 2022).

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
