# Peer review of "NAGS*, *CPS1*, and *SLC25A13* (Citrin) at the Crossroads of Arginine and Pyrimidines Metabolism in Tumor Cells"

_ijms, 2023, doi:10.3390/ijms24076754_

Round 1
Reviewer 1 Report
The present study assessed the expression pattern of urea cycle protein including NAGS, CPSI and citrin. though the data presented here are based on data mining without any experimental evidence, the study is novel and interesting and open more avenue for further studies. My only comment is the resolution of some figures especially figure 3 is low and should be improved and the paper should be checked carefully for typos and errors particularly for abbreviation
Author Response
Reviewers’ comments have been formatted in italics, and our responses are in the regular blue typeface.
The present study assessed the expression pattern of urea cycle protein including NAGS, CPSI and citrin. though the data presented here are based on data mining without any experimental evidence, the study is novel and interesting and open more avenue for further studies. My only comment is the resolution of some figures especially figure 3 is low and should be improved and the paper should be checked carefully for typos and errors particularly for abbreviation
All figures have been re-imported into template manuscript file as either PDF or SVG files that contain vector graphics. This should allow image resizing without loss of resolution.

Reviewer 2 Report
This paper addresses the question of the role of ectopically expressed urea cycle (UC) enzymes in cancer. It examines public databases to infer from the deposited data whether there are changes in the expression of these enzymes in a number of tumors for which transcriptomic and proteomic data do exist. The paper not only examines the existing data on the expression of urea cycle enzymes in tumors as transcripts, but it also examines for three UC proteins, NAGS, CPS1 and citrin, the database records on levels of transcripts and proteins (from a proteomic approach) in several tissues, to infer if transcriptomic data can be considered a proxy for protein levels, concluding that they are a good proxy. Then they examine in a number of tumors in which they find increased expression of these three proteins whether there is or there is not a relation between their levels and severity. They then examine reasons for increased transcription by searching for epigenetic changes at regulatory sites for expression of he encoding genes, also looking for mutations therein, and finally for copy number changes for each gene. Unrelated to mRNA levels but related to activity and stability of the produced proteins, they also examine the occurrence of missense mutations in the corresponding genes in the tumors, applying a predictive tool to estimate the degree of derangement of the protein because of these muations. I do not find very strong novel conclusions, but even if it were only for gathering, ordering and presening the data, the work is worthy of publication, more so since some inferences are made and alternative hypotheses considered that make the discussion interesting and enlightening.
That said, there are is number of issues that must be considered and fixed.
1. Lines 56-57, sentence “Aberrant expression of enzymes in catabolic pathways such as urea cycle can contribute to”. I think that this formulation is misleading. Please change to “Aberrant expression of enzymes belonging to catabolic pathways, such as the urea cycle, can contribute to ….
2. Lines 25-26 “ Expression of the NAGS gene and its product was not evaluated in tumors with aberrant expression of CPS1 and citrin”; lines 95-97 “….none examined whether these tumor cells also express NAGS which is needed for production of NAG that is an obligate allosteric activator of CPS1 [32–35].” Although it is true that NAG levels have not been measured, the enzyme has been proven to be highly expressed in three cell lines of non-small cell lung cancer (see Makris G, Kayhan S, et al. Impact of small molecule-mediated inhibition of ammonia detoxification on lung malignancies and liver metabolism. Cancer Commun (Lond). 2023 Jan 28., last paragraph of page 3). It is also true that cell lines derived from tumors are not tumors, but,nevertheless, please take into accont this information in the introduction and discuss your results considering in the discussion this published finding. Perhaps this should b considered, too, in the last two lines of the Abstract, and in the part of the discussion on the alternatives for CPS1 activation in the tumors.
3. This paper is intended for a general biochemical audience, not for one speciallized in bioinformatics or on gene expression. Therefore, all should be comprehensible without having to go to the original databases. The lack of reader-friendly language is patent from the the first lines of the Results, when you write “We queried the GDAC Firehose Data Portal for mRNA expression of urea cycle genes in tumors and surrounding normal tissue. The difference in expression of urea cycle genes between tumor and surrounding normal tissue, expressed as fold change of median log2RSEM values, was available for 28 tumor types (Figure 1A).”
a) Please indicate what is a RSEM value by defining it or by referencing to the original paper where the RSEM system was described.
b) In addition, the significance of medians strongly depends on the number of samples from which the median was derived. Please give some hint on the number of samples used for median estimation (for example, “n > x for all tumors considered here”).
c) Do you have for each individual tumor sample the corresponding value for the surrounding tissue of the individual from whom the tumor was removed? If so, how do you process it? Do you estimate for each sample the quotient resulting from dividing the value in the tumor by the value in the surrounding tissue? Do you then order the quotients obtained for all the samples, estimating the median? Please explain this process in detail (perhaps in the Methods), since this part is not easy to understand. In fact, it does not help your use of the expression “The difference in expression…..between tumors and surrounding tissue” because a difference is a subtraction, and I don’t think this could be what you did.
4. In Fig. 1, please spell out TPM in the figure itself and define IBAQ in he legend.
5. The very small values in the cholangiocarcinoma (Fig. 1, associated text and Supplementary table 1) must reflect the fact that this is a cancer of the biliary system (tubes that collect the bilis) and that this system is in the liver, where the hepatocytes are very high in urea cycle enzymes. It makes no sense to consider the liver as the surrounding tissue because you are interested on the ratio between expression of urea cycle enzymes in the tumor and the expression (if any) in the same tissue (the biliary tract cells) when not tumoral. As this may be impossible, I would eliminate this tumor from Fig. 1 and from the text.
6. Your denomination of tumors is quite confusing. In particular the use of glyoblastome through the text is very very confusing. For clarity, please adhere strictly to the naming and abbreviations of the tumors given in the legend to Fig. 1, and give always the abbreviation too (perhaps in parentheses next to the name). This applies to text and figures.
7. I am not an expert on which is the function of aralar, the paralogous gene product that is expressed predominantly in most tissues of the body excluding the liver, but I see no reason why, catalyzing the same exchanges across the mitochondrial membrane, it should not non-be involved in the shuttle in other tissues, like the muscle or the heart. Therefore, I think that you should not rely on changes in operation of this shuttle due to changes in citrin expression in tissues in which aralar is abundantly expressed. Please revise the paper keeping in mind the aralar expression data for non-hepatic tissue, to judge if an increase in the expression of cirin in these tissues could have any metabolic impact.
8. It might appear surprising that NAGS makes a change in glyoblastoma, as mammalian brain tissue has so much NAG as liver has (per gr of each organ, see Neurochem Res 1991;16:787). However, the NAG fond in the brain (which is not produced by NAGS) is cytosolic instead of mitochondrial. Perhaps it is worth mentioning this in your discussion.
9. Please add legends to he supplementary tables, so that they may be understood more easily. The legends should give the units used.
Author Response
Reviewers’ comments have been formatted in italicsand our responses are in regular blue typeface.
Lines 56-57, sentence “Aberrant expression of enzymes in catabolic pathways such as urea cycle can contribute to”. I think that this formulation is misleading. Please change to “Aberrant expression of enzymes belonging to catabolic pathways, such as the urea cycle, can contribute to ….
The text has been changed as suggested by the reviewer.
Lines 25-26 “ Expression of the NAGS gene and its product was not evaluated in tumors with aberrant expression of CPS1 and citrin”; lines 95-97 “….none examined whether these tumor cells also express NAGS which is needed for production of NAG that is an obligate allosteric activator of CPS1 [32–35].” Although it is true that NAG levels have not been measured, the enzyme has been proven to be highly expressed in three cell lines of non-small cell lung cancer (see Makris G, Kayhan S, et al. Impact of small molecule-mediated inhibition of ammonia detoxification on lung malignancies and liver metabolism. Cancer Commun (Lond). 2023 Jan 28., last paragraph of page 3). It is also true that cell lines derived from tumors are not tumors, but,nevertheless, please take into accont this information in the introduction and discuss your results considering in the discussion this published finding. Perhaps this should b considered, too, in the last two lines of the Abstract, and in the part of the discussion on the alternatives for CPS1 activation in the tumors.
We are happy to include results published by Makris et al. while our manuscript has been under review. The abstract (lines 26-27) has been modified as follows: “Although NAGS is expressed in lung cancer derived cell lines, expression of the NAGS gene and its product was not evaluated in tumors with aberrant expression of CPS1 and citrin.” The introduction (lines 94-97) has been modified as follows: Although these studies provided evidence of CPS1 enzymatic activity in tumor cells, only one study showed presence of NAGS in cell lines that model non-small cell lung carcinoma and express CPS1.” The discussion (lines 623-626) has been modified as follows: ”NAGS is expressed in two cell lines commonly used to model non-small cell lung carcinoma [32]; this provides a molecular mechanism for CPS1 activity and increased production of CP in the two cell lines.” The last paragraph of the discussion was also revised to include information about AT067-H09 inhibitor of CPS1.
This paper is intended for a general biochemical audience, not for one speciallized in bioinformatics or on gene expression. Therefore, all should be comprehensible without having to go to the original databases. The lack of reader-friendly language is patent from the the first lines of the Results, when you write “We queried the GDAC Firehose Data Portal for mRNA expression of urea cycle genes in tumors and surrounding normal tissue. The difference in expression of urea cycle genes between tumor and surrounding normal tissue, expressed as fold change of median log2RSEM values, was available for 28 tumor types (Figure 1A).”
Description of the GDAC Firehose data portal has been added to Methods (110-111) and Results (221-223) sections. Results also include a statement that “GDAC uses the Log2 transform of RSEM values [69] to quantify abundance of mRNA.” (225-226).
a) Please indicate what is a RSEM value by defining it or by referencing to the original paper where the RSEM system was described.
The use of RSEM for quantification of transcript abundance from RNA-seq data has been developed by Li et al. (PMID: 21816040). This publication has been added as a reference in Methods and Results sections. RSEM (RNA-seq Expectation-Maximization) values are calculated from the number of sequencing reads using statistical methodology that accounts for the possibility of sequencing reads aligning to multiple transcripts.
b) In addition, the significance of medians strongly depends on the number of samples from which the median was derived. Please give some hint on the number of samples used for median estimation (for example, “n > x for all tumors considered here”).
The number of tumor and matched normal tissue samples used to calculate the median expression values has been added to as a table in the Supporting File 1 and Figure 1 legend (lines 240-241). The number of glioblastoma multiforme, glioma, stomach adenocarcinoma, stomach and esophageal carcinoma and lung adenocarcinoma samples used to calculate the fold-change in expression of urea cycle genes has been added to results section (lines 232-236).
c) Do you have for each individual tumor sample the corresponding value for the surrounding tissue of the individual from whom the tumor was removed? If so, how do you process it? Do you estimate for each sample the quotient resulting from dividing the value in the tumor by the value in the surrounding tissue? Do you then order the quotients obtained for all the samples, estimating the median? Please explain this process in detail (perhaps in the Methods), since this part is not easy to understand. In fact, it does not help your use of the expression “The difference in expression…..between tumors and surrounding tissue” because a difference is a subtraction, and I don’t think this could be what you did.
The median Log2RSEM values for each urea cycle gene in tumor and matched normal tissue samples were determined by GDAC. For each urea cycle gene in each tumor type, median Log2RSEM value for the matched normal tissue samples was subtracted from the median Log2RSEM value in tumor samples. The fold change values were then calculated as 2 raised to the power of the difference between the median Log2RSEM values in tumor samples and matched normal tissues. This text has been added to the Methods section (lines 114-121).
In Fig. 1, please spell out TPM in the figure itself and define IBAQ in he legend.
The y-axes labels in Figure 1C, 1F and 1I were changed to show TPM as a unit of mRNA abundance. The y-axes labels in Figure 1E, 1H and 1K were changed to show iBAQ as a unit of protein abundance. Definitions of TPM and iBAQ were added to Figure 1 legend (lines 247-248 and 251-252, respectively).
The very small values in the cholangiocarcinoma (Fig. 1, associated text and Supplementary table 1) must reflect the fact that this is a cancer of the biliary system (tubes that collect the bilis) and that this system is in the liver, where the hepatocytes are very high in urea cycle enzymes. It makes no sense to consider the liver as the surrounding tissue because you are interested on the ratio between expression of urea cycle enzymes in the tumor and the expression (if any) in the same tissue (the biliary tract cells) when not tumoral. As this may be impossible, I would eliminate this tumor from Fig. 1 and from the text.
Cholangiocarcinoma data were removed from Figures 1A-B and from the text as suggested by the reviewer.
Your denomination of tumors is quite confusing. In particular the use of glyoblastome through the text is very very confusing. For clarity, please adhere strictly to the naming and abbreviations of the tumors given in the legend to Fig. 1, and give always the abbreviation too (perhaps in parentheses next to the name). This applies to text and figures.
Abbreviations for glioblastoma multiforme (GBM), glioma (GBMLGG), lung adenocarcinoma (LUAD), stomach adenocarcinoma (STAD), and stomach and esophageal carcinoma (STES) remain in the text describing Figures 1A-B because the same abbreviations are used in the figure. The abbreviations also remain in the abstract for brevity. In the rest of the manuscript abbreviations for different tumor types have been replaced with tumor names to be consistent with figures and figure legends.
I am not an expert on which is the function of aralar, the paralogous gene product that is expressed predominantly in most tissues of the body excluding the liver, but I see no reason why, catalyzing the same exchanges across the mitochondrial membrane, it should not non-be involved in the shuttle in other tissues, like the muscle or the heart. Therefore, I think that you should not rely on changes in operation of this shuttle due to changes in citrin expression in tissues in which aralar is abundantly expressed. Please revise the paper keeping in mind the aralar expression data for non-hepatic tissue, to judge if an increase in the expression of cirin in these tissues could have any metabolic impact.
ARALAR1 is a citrin paralog with identical biochemical function (PMID: 27132995); ARALAR1 is highly expressed in the brain, skeletal muscle and heart, present in other tissues and absent from the liver where citrin is highly expressed (PMID: 27132995). Increased supply of cytoplasmic aspartate due to ectopic overexpression of citrin could result in higher CAD activity, higher de novo pyrimidine biosynthesis and metabolic reprogramming that leads to increased cell proliferation. This text has been added to discussion (lines 632-637).
It might appear surprising that NAGS makes a change in glyoblastoma, as mammalian brain tissue has so much NAG as liver has (per gr of each organ, see Neurochem Res 1991;16:787). However, the NAG fond in the brain (which is not produced by NAGS) is cytosolic instead of mitochondrial. Perhaps it is worth mentioning this in your discussion.
Aberrant overexpression of NAGS could contribute to poor outcomes for patients with glioblastoma and glioblastoma multiforme independently of CPS1. NAG has been detected in the cytoplasm of mammalian and avian brain cells (PMID: 1944768, 4964109) but the function of NAG in the brain remains poorly understood. Abundance of NAG in mammalian brain and liver are similar (PMID: 1944768). NAG that is produced in the liver mitochondria is transported to the cytoplasm for degradation (PMID: 902767). A similar transport mechanism may operate in the brain cells allowing NAG, produced in the mitochondria of brain cells by the aberrantly overexpressed NAGS, to be transported in the cytoplasm, increase the pool of cytoplasmic NAG and contribute to metabolic reprogramming and tumorigenesis. This text has been added to discussion (lines 703-711).
Please add legends to the supplementary tables, so that they may be understood more easily. The legends should give the units used.
Supplementary Tables have been modified as suggested by the reviewer.
